# D-Dimer Levels Are Not Elevated in SARS-CoV-2 IgG Positive Patients Undergoing Elective Orthopedic Surgery

**DOI:** 10.3390/jcm10163508

**Published:** 2021-08-09

**Authors:** Anna Jungwirth-Weinberger, Lisa Oezel, Rachelle Morgenstern, Jennifer Shue, Carola Hanreich, Andrew A. Sama, Friedrich Boettner

**Affiliations:** 1Adult Reconstruction and Joint Replacement Service, Hospital for Special Surgery, 535 East 70th Street, New York, NY 10021, USA or anna.jungwirth-weinberger@ksb.ch (A.J.-W.); morgensternr@hss.edu (R.M.); hanreichc@hss.edu (C.H.); 2Orthopaedics and Traumatology, Cantonal Hospital Baden, Im Ergel 1, CH-5404 Baden, Switzerland; 3Spine Care Institute, Hospital for Special Surgery, 535 East 70th Street, New York, NY 10021, USA; oezell@hss.edu (L.O.); shuej@hss.edu (J.S.); samaa@hss.edu (A.A.S.)

**Keywords:** COVID-19, SARS-CoV-2 IgG, D-Dimer, spine surgery, total knee replacement, total hip replacement

## Abstract

Introduction: In acute COVID-19, D-Dimer levels can be elevated and those patients are at risk for thromboembolic events. This study aims to investigate differences in preoperative D-Dimer levels in SARS-CoV-2 IgG positive and negative patients undergoing primary total knee and total hip replacement (TJA) or spine surgery. Methods: D-Dimer levels of 48 SARS-CoV-2 IgG positive and 718 SARS-CoV-2 IgG negative spine surgery patients were compared to those of 249 SARS-CoV-2 IgG positive and 2102 SARS-CoV-2 IgG negative TJA patients. Patients were assigned into groups based on D-Dimer levels as follows: <200 ng/mL, 200–400 ng/mL, and >400 ng/mL D-Dimer Units (DDU). Results: D-Dimer levels did neither differ significantly between SARS-CoV-2 IgG positive spine surgery patients and TJA patients (*p* = 0.1), nor between SARS-CoV-2 IgG negative spine surgery and TJA patients (*p* = 0.7). In addition, there was no difference between SARS-CoV-2 IgG positive and negative spine surgery patients and SARS-CoV-2 IgG positive and negative TJA patients (*p* = 0.3). Conclusions: There is no difference in D-Dimer levels between SARS-CoV-2 IgG positive and negative patients and there does not seem to be any difference for different orthopedic specialty patients. Routine testing of D-Dimer levels is not recommended for patients undergoing elective orthopedic surgery.

## 1. Introduction

COVID-19 infection is caused by the severe acute respiratory syndrome coronavirus 2 (SARS-CoV-2) [1,2]. The first cases presented in Wuhan, China, and were associated with the Wuhan Wet market and it is assumed the virus has animal origin. SARS-related coronaviruses are covered by spike proteins that contain a receptor-binding domain, which bind to the angiotensin converting enzyme-2 (ACE-2) in humans [3]. Transmission between human mostly occurs via person-to-person route through respiratory droplets [4].

Three stages of the disease have been defined: stage I (mild symptoms), stage II (pulmonary involvement) and stage III (systemic inflammation). Hematologic symptoms include alterations in the white blood cell (WBC) count, low lymphocyte count, a low platelet count and elevation of D-Dimer associated with high levels of fibrin degradation products (FDPs) and low antithrombin (AT) activity [5].

D-Dimer levels might be elevated in active COVID-19 [6,7] and those patients are at risk for thromboembolic events both in the arterial as well as the venous system [8,9]. Severe elevation is associated with a worse outcome and elevated D-Dimer levels represent a poor prognostic factor [10,11]. D-Dimer is a fibrin degradation product and elevated D-Dimer levels can be observed during deep vein thrombosis (DVT) and pulmonary embolism (PE), but also in other clinical conditions such as injuries, after surgery, during infection and inflammation or cancer [12,13,14,15,16].

DVT and PE are serious postoperative complications after orthopedic surgeries and spine surgeries and lead to a significant increase in mortality [17].

D-Dimer elevation in arthroplasty patients is a moderately sensitive, but less specific marker in the detection of DVT after total knee arthroplasty (TKA) [18], and elevated D-Dimer represents a risk factor for DVT in TKA patients [19]. Some authors use D-Dimer as screening test to identify DVT after orthopedic surgery and sensitivity are reported to be 71.4% and 81.7%, respectively [20]. In combination with blood gas analysis, D-Dimer can be utilized as screening tool for pulmonary embolism after orthopedic or spinal surgery [21].

D-Dimer has also been shown to be a marker for periprosthetic infection due to a correlation between coagulation and inflammation: coagulation-related biomarkers have a proinflammatory effect and persistent inflammatory response contributes to a hyper-coagulable state which leads to elevation of D-Dimer in infectious disease [22,23,24,25]. In contrast, D-Dimer levels cannot be used to determine the timing for reimplantation [26].

In neurosurgery patients, higher D-Dimer levels are associated with an increased risk of non-routine discharge [27]. Elevated D-Dimer levels have been shown to be a predictive factor for postoperative PE after spine surgery [28,29,30]. A D-Dimer level of ≥5000 ng/mL was defined to be a risk factor fort thromboembolic events after spinal surgery, although false positive cases can occur [31].

The impact of previous SARS-CoV-2 infection on the morbidity of patients undergoing elective orthopedic surgery has not yet been assessed. We hypothesized that D-Dimer levels in patients with history of SARS-CoV-2 are higher in both arthroplasty and spine patients.

This study aims to evaluate D-Dimer levels in TJA patients and spine surgery patients and analyze for potential differences between SARS-CoV-2 IgG positive and negative patients.

## 2. Materials and Methods

This retrospective study was approved by the local institutional review board.

Between June and October 2020 D-Dimer levels were prospectively collected and retrospectively reviewed in every elective arthroplasty patient and between June and August 2020 in every patient undergoing spine surgery at the authors’ institution. In SARS-CoV-2 IgG positive arthroplasty patients, D-Dimer levels were drawn between June and October 2020 as per the policy of the Adult Reconstruction and Joint Replacement service.

D-Dimer levels were drawn during presurgical screening 3–21 days (average 7 days) prior to surgery. D-Dimer levels were reported semi-quantitively by the institution and were categorized as follows: normal values were <200 ng/mL D-Dimer Units (DDU), elevated D-Dimer levels were classified as data within the ranges from 200–400 ng/mL, and over 400 ng/mL DDU. The polymerase chain reaction (PCR) test (Cepheid Xpert Xpress SARS-CoV-2 [32] (Sunnyvale, CA, USA; positive percent agreement 98%, negative percent agreement 96%) or the Biomerieux Biofire Respiratory Panel 2.1 [33] (Salt Lake City, UT, USA; positive percent agreement 98% and negative percent agreement 100%) for active COVID-19 infection was negative in all patients at the day of surgery and SARS-CoV-IgG-2 positive patients only had surgery, if they recovered from COVID-19 and subsequently tested negative on PCR test prior to their surgery. The serological SARS-CoV-2 IgG status (Abbott Architect, IgG sensitivity 100%, specificity 99%; Abbott Park, IL, USA [34]) was determined during presurgical screening 3–21 days prior to surgery in all patients.

The TJA group consisted of 2351 patients, 249 were SARS-CoV-2 IgG positive and 2102 SARS-CoV-2 IgG negative. The spine group consisted of 766 patients undergoing surgery for degenerative spine conditions and spinal stenosis, 48 were SARS-CoV-2 IgG positive and 718 SARS-CoV-2 IgG negative.

Only patients undergoing elective, non-septic surgeries were included and periprosthetic infections, incision and drainage as well as surgeries for spondylodiscitis were excluded in both groups.

The primary choice DVT prophylaxis in TJA patients was acetylsalicylic acid for four-six weeks compared to low molecular heparin during hospital stay in spine patients.

Analyses were conducted using SAS software version 9.4 (SAS Institute Inc., Carey, NC, USA). Variables were assessed for normalcy; comparisons between and within cohorts were made using Wilcoxon rank-sum tests and Wilcoxon signed-rank tests, respectively. Categorical variables were assessed using Chi-square or Fisher exact tests.

## 3. Results

249 patients in the TJA group were SARS-CoV-2 IgG positive (146 female, 103 male) and 2102 were SARS-CoV-2 IgG negative (1229 female, 873 male).

In the SARS-CoV-2 IgG positive TJA group the mean age was 62.9 years (range 18–86 years), mean BMI was 30.9 kg/m^2^ (16.8–49.9 kg/m^2^). Mean age in SARS-CoV-2 IgG negative TJA patients was 65.3 years (18–96 years), mean BMI was 29.5 kg/m^2^ (15.6–56.8 kg/m^2^).

The spine group consisted of 48 SARS-CoV-2 IgG positive patients (34 male, 14 female) and 718 SARS-CoV-2 IgG negative patients (313 female, 405 male).

Mean age in the SARS-CoV-2 IgG positive spine group was 55.4 years (21–78 years), mean BMI was 29.6 kg/m^2^ (19–44.6 kg/m^2^). Amongst SARS-CoV-2 IgG negative patients, mean age was 60.4 years (19–91 years), mean BMI was 28.4 kg/m^2^ (17.4–50.9 kg/m^2^) (Table 1).

Average interval between COVID-19 and surgery was 168 days (SD 71).

When comparing SARS-CoV-2 IgG positive spine and TJA patients, spine patients were significantly younger (*p*-value < 0.0001). BMI was not significantly different between the two groups (*p* = 0.2).

SARS-CoV-2 IgG positive patients were significantly younger (61.7 years, SD 11.7, range 18–86 years, *p* = 0.007) and had a higher BMI (30.7 kg/m^2^, SD 6.1, range 16.8–49.9, *p* < 0.0001) than SARS-CoV-2 IgG negative patients (64.1 years, SD 11.6, 19–96 years; 29.1 kg/m^2^, SD 5.8, 15.6–56.8 kg/m^2^).

A total of 197 patients (79.1%) of the TJA SARS-CoV-2 IgG positive patients had D-Dimer levels <200 ng/mL, 36 patients (14.5%) between 200–400 ng/mL and 17 patients (6.43%) >400 ng/mL. A total of 38 patients (79.2%) of the spine SARS-CoV-IgG-2 positive patients had D-Dimer levels <200 ng/mL, 10 patients (20.8%) between 200–400 ng/mL. The difference in D-Dimer levels between the two SARS-CoV-2 IgG positive groups was not significantly different (*p*-value = 0.1).

Among SARS-CoV-2 IgG negative TJA patients, 1709 patients (81.3%) had D-Dimer levels <200 ng/mL, 255 patients (12.1%) between 200–400 ng/mL and 138 patients (6.6%) >400 ng/mL. In the SARS-CoV-2 IgG negative spine group 587 patients (81.8%) had a D-Dimer level <200 ng/mL, 90 patients (12.5%) between 200–400 ng/mL and 41 patients (5.7%) >400 ng/mL. The difference was not significantly different between those two groups (*p*-value = 0.7).

Comparing spine patients to TJA patients, regardless of their SARS-CoV-2 IgG status, the difference between D-Dimer levels was not significantly different (*p* = 0.5).

Comparing only the SARS-CoV-2 IgG positive spine and TJA patients to SARS-CoV-2 IgG negative spine and TJA patients, there was also no significant difference between the groups (*p* = 0.24) (Table 2).

No DVTs or PEs occurred in the spine group, but there was one symptomatic PE (0.4%) and one symptomatic DVT (0.4%) in SARS-CoV-2 IgG positive TJA patients.

## 4. Discussion

The current study does not show a difference in preoperative D-Dimer levels between SARS-CoV-2 IgG positive and negative spine and TJA patients.

Routine testing of D-Dimer before elective orthopedic surgery in clinical practice is not supported by this.

Another finding of the study is, that there was no difference of D-Dimer levels between spine and TJA patients, regardless of their SARS-CoV-2 IgG status.

To our knowledge, this is the first study comparing preoperative D-Dimer levels for different orthopedic indications. Therefore, this study indicates, that osteoarthritis does not lead to increased D-Dimer levels in TJA patients compared to disc herniation or spinal stenosis in spine patients.

Contrarily, a study by Cheras et al. [35] has been shown D-Dimer elevation in osteoarthritis due to a hypercoagulable and prothrombotic condition in osteoarthritis. Patients with rheumatoid arthritis showed higher D-Dimer values prior to total knee arthroplasty than osteoarthritis patients and those values remained elevated for a week postoperatively [36]. Postoperative ambulation had a strong correlation with D-Dimer levels in a study by Nakao et al. Non-ambulatory patients had significantly higher D-Dimer levels than ambulatory patients [37] and also the preoperative ambulatory ability might influence postoperative D-Dimer levels in patients undergoing total hip arthroplasty [38].

D-Dimer can be used as prognostic factor in different tumors [39,40] and was recently found as marker for periprosthetic infection in arthroplasty patients [23,24]. A meta-analysis including 1592 patients showed a sensitivity and specificity of D-Dimer for diagnosing a periprosthetic joint infection of 82% and 73%, respectively [41], and D-Dimer levels were significantly higher in patients with periprosthetic joint infection than in patients with aseptic failure [42]. However, other studies found no significant difference in D-Dimer levels between septic and aseptic total hip and knee arthroplasties [43,44].

Although D-Dimers have been used for some time for the detection of DVT and PE, it has been recently discussed controversial due to its low specificity [18]. A study by Rafee et al. found no difference in postoperative D-Dimer levels in patients with and without DVT [45]. Niimi et al. recommend due to D-Dimer’s low specificity a two-stage screening for DVT, first with D-Dimer or soluble fibrin, followed by venography or sonography [46]. In clinical practice, D-Dimer levels are most commonly utilized to exclude a DVT, although it should not be used as stand-alone test due to its low specificity. The sensitivity of D-Dimer is >98%. Recently D-Dimer has also been used to decide whether anticoagulation in DVT can be terminated or not [47], and the risk of recurrence of a DVT is significantly lower in patients with normal D-Dimer values measured one month after discontinuation of anticoagulation compared to patients with elevated D-Dimer levels [48].

D-Dimer has been shown to rapidly rise and fall after TJA with a peak on postoperative day one and returns to normal values after six weeks [49]. At postoperative day two, D-Dimer normally decreases to its baseline level and slowly increases until postoperative week two [49].

In acute COVID-19, D-Dimer has been shown to be elevated and [50] COVID-associated coagulopathy with evidence of microthrombi and macrothrombi in the venous and arterial systems can be diagnosed based on D-dimer elevations [51]. In addition, death, intubation and thromboembolic events were associated with admission D-Dimer [52]. D-Dimer levels >3000 ng/mL were associated with pulmonary embolism in a retrospective study of 88 patients hospitalized for COVID-19 and the authors defined D-Dimer as independent risk factor of PE during COVID-19 [53]. Non-Survivors of COVID-19 revealed significantly higher D-Dimer levels and fibrin degradation products on admission than survivors of COVID-19 [54]. Zhou et al. report higher D-Dimer levels in patients admitted to the ICU compared to non-ICU patients [55]. Elevated D-Dimer levels were shown in 18.8% of all our patients, therefore we assume, the bias of postponed surgeries due to elevated D-Dimer levels has been eliminated.

Limitations of the study are (1) there was no follow up during postoperative course and no routine DVT screening; (2) the current study focuses on an early period following the initial COVID-19 outbreak, with a less than 6 months interval between infection and spine or TJA surgery. It is likely that patients with a severe course and hospitalization did not schedule elective surgery during this early time period. (3) A post hoc power analysis for cohorts with non-significant *p*-values usually result in low power and is therefore not recommended. The large sample size of 3118 patients including 297 COVID positive patients should allow for a meaningful conclusion. A power-analysis was not performed a priori because preliminary data for establishing an effect size in this setting were not available.

## 5. Conclusions

There is no difference in D-Dimer levels between SARS-CoV-2 IgG negative and SARS-CoV-2 IgG positive patients undergoing TJA and spine surgery and no difference in D-Dimer levels between TJA and spine patients. The current study does not support routine D-Dimer testing in SARS-CoV-2 IgG positive patients prior to elective TJA or spine surgery.

## Figures and Tables

**Table 1 jcm-10-03508-t001:** Demographic information of the groups.

	SARS-CoV-2 IgG positive patients		
	TJA Patients	Spine Patients	*p*-Value
*n*	249	48	
Male/Female	103/146	34/14	0.0002 *
Age (years)	62.9 (SD 10.3; 18–86)	55.4 (SD 15.8; 21–78)	<0.0001 **
BMI (kg/m^2^)	30.8 (SD 6.2; 16.8–49.9)	29.6 (SD 5.8; 19–44.6)	0.2 **
	SARS-CoV-2 IgG negative patients		
*n*	2109	718	
Male/Female	873/1229	405/313	<0.0001 *
Age (years)	65.3 (SD 10.7; 19–96)	60.4 (SD 14.2; 19–91)	<0.0001 **
BMI (kg/m^2^)	29.5 (SD 6.0 (15.6–56.8)	28.4 (SD, 5.6; 17.4–50.9)	<0.0001 **

* Chi-square-Test; ** *t*-Test.

**Table 2 jcm-10-03508-t002:** Distribution of D-Dimer levels of SARS-CoV-2 IgG positive and negative patients.

	D-Dimer Levels (DDU)			*p*-Value
	<200 ng/mL	200–400 ng/mL	>400 ng/mL	
SARS-CoV-2 IgG positive	235 (79.1%)	46 (15.5%)	16 (5.4%)	0.24 *
- TJA	197	36	16	
- Spine	38	10	0	
SARS-CoV-2 IgG negative	2296 (81.4%)	345 (12.2%)	179 (6.4%)	
- TJA	1709	255	138	
- Spine	587	90	41	

* Chi-square-Test.

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
