# Peer review of "D-Dimer Levels Are Not Elevated in SARS-CoV-2 IgG Positive Patients Undergoing Elective Orthopedic Surgery"

_jcm, 2021, doi:10.3390/jcm10163508_

Round 1
Reviewer 1 Report
The authors did not revise the manuscript as recommended by the reviewers, and the previous comments were not taken into account. The manuscript contains many methodological errors indicated in numerous comments from many reviewers.
Author Response
We are very sorry the manuscript was not revised to your satisfaction. Thebackground of this study is the first wave in New York City, which hit the city very hard and resulted in a high percentage of the population contracting COVID-19. When initiating a return to elective orthopedic surgery, there was considerable concern that patients with a history of COVID-19 might be at increased risks for thromboembolic events. With more than 30 Mio COVID-19 cases in the US alone this could have significant impact on elective orthopedic surgeries throughout the country.
The large sample size of 3118 patients including 297 COVID positive patients should allow for a meaningful conclusion. A power-analysis was not performed a priori because preliminary data for establishing an effect size in this setting were not available. A post hoc power analysis for cohorts with non-significant p-values usually result in low power and is therefore not recommended.
Reviewer 2 Report
Jungwirth-Weinberger et al. report D-Dimer levels in some SARS-CoV-2 IgG positive patients and compare these levels between TJA and Spine patients.
Power analysis: Please report the fact that your study is underpowered. It is important for the reader to understand that due to the low number of patients included, the clinical meaningfulness is limited.
The overall purpose of this study is still not evident to me. You state that it is important to assess D-Dimer levels in patients with "history of SARS-CoV-2" undergoing arthroplasty and spine patients. But you admit to be underpowered. So why is it important to know the increased D-Dimer levels in these patients at all? Especially, if there is no direct comparison between SARS-CoV-2 IgG positive vs. negative patients in table- or figure format.
Please report which test was used for which comparison within the tables or results section.
Tab. 1: You compare TJA patients to Spine patients. This does not make sense. As i suggested earlier, please compare SARS-CoV-2 positive vs. negative patients. If you choose to, you can compare them within the TJA group and within the Spine group. But this is the important comparison, not TJA vs. Spine patients.
No, Table 1 does not provide the comparison I asked for.
Without proper coagulation testing, the statement "This is an important finding that suggests, that patients with 170 history of COVID-19 infection are not in a hypercoagulable state throughout the duration of their antibody positivity." cannot stand. Either provide more sophisticated measurements or delete the sentence.
Author Response
Thank you for your valuable comments.
We added a sentence to the result section that the study is underpowered.
COVID-19 is associated with thromboembolic events during the acute infection, to assess the risk for patients undergoing elective surgery, D-dimers were peroperatively determined to identify patients with a persistent hypercoagulable state. In regard to the power calculation we would like to point out that we are not reporting on detecting actual DVT rates. To identify DVT this the study is underpowered, however, the current paper focuses on D-dimer levels, using them as indicator for a potentially increased risk for thromboembolic events. The large sample size of 3118 patients including 297 COVID positive patients should allow for a meaningful conclusion. A power-analysis was not performed a priori because preliminary data for establishing an effect size in this setting were not available. A post hoc power analysis for cohorts with non-significant p-values usually result in low power and is therefore not recommended.
We added the used tests to tables.
Table 2 was changed to a comparison of D-Dimer levels of SARS-CoV-2 IgG positive and negative patients.
Reviewer 3 Report
The revision definitely improved the quality of the paper.
Author Response
Thank you for your kind comment.
Round 2
Reviewer 1 Report
The literature used in the manuscript is very scarce and should be expanded to include the following articles:
oi: 10.3390/pathogens9030231,
doi: 10.3390/pathogens9060493,
doi: 10.3390/pathogens9070519.
Author Response
Thank you for your valuable comment, we added those references and incorporated them in the introduction.Reviewer 2 Report
I want to thank the authors for improving their manuscript.
However, I still have serious concerns with the study. There are multiple studies suggesting the need for assessment of d-dimers in this patient cohort, and even pointing out an excess risk of VTE in COVID-19 patients (PMID: 32581015,PMID: 32412320, PMID: 33995770, PMID: 33534149). Even if your cohort is just "SARS-CoV-2 IgG postive", not necessarily in an acute SARS-CoV-2 state, the lack of follow-up prevents your results to be clinically meaningful.
Author Response
Thank you for your comment.
This manuscript does not deal with patients with an active COVID-19 disease, but with patients who have had COVID-19 in the past and recovered from the disease. With this study, we do not want to state that determination of D-dimer is not indicated in active Covid-19 disease, but we have concluded at our institution that D-Dimer levels in patients undergoing elective arthroplasty or spine surgery do not seem necessary because there are no differences in the levels to patients without a history of COVID-19.
All the studies you cite deal with patients with active, severe Covid-19 disease. All our patients were tested prior to surgery and had a negative PCR test, so none of the patients had active COVID disease.
Guevara-Noriega et al. describe a derangement of coagulation function in patients with SARS-CoV2 infection and find that D-Dimer and FDP are the most significantly altered values which also correlate with severity.
Giorgi et al operate on COVID-19 patients with fractures. This is not comparable to our group of elective PCR negative patients.
Li et al report a symptomatic VTE rate of 5.94% in patients hospitalized for severe COVID-19 and 2.79% in non-severe hospitalized COVID-19 cases and compare them with a historical cohort of inpatients for medical conditions, whereas undergoing surgery was an exclusion criterion in this cohort.
- Guevara-Noriega KA, Lucar-Lopez GA, Nuñez G, Rivera-Aguasvivas L, Chauhan I. Coagulation Panel in Patients with SARS-CoV2 Infection (COVID-19). Ann Clin Lab Sci. 2020 May;50(3):295-298. PMID: 32581015.
- Giorgi PD, Gallazzi E, Capitani P, Biancardi E, Bove F, Mezzadri U, Capitani D, Schirò GR. Mortality and morbidity in COVID-19 orthopedic trauma patients: is early surgery the keystone? Pan Afr Med J. 2021 Feb 12;38:163. doi: 10.11604/pamj.2021.38.163.27125. PMID: 33995770; PMCID: PMC8077645.
- Li JY, Wang HF, Yin P, Li D, Wang DL, Peng P, Wang WH, Wang L, Yuan XW, Xie JY, Zhou F, Xiong N, Shao F, Wang CX, Tong X, Ye H, Wan WJ, Liu BD, Li WZ, Li Q, Tang LV, Hu Y, Lip GYH; Thrombo-COVID-19 Collaborative. Clinical characteristics and risk factors for symptomatic venous thromboembolism in hospitalized COVID-19 patients: A multicenter retrospective study. J Thromb Haemost. 2021 Apr;19(4):1038-1048. doi: 10.1111/jth.15261. Epub 2021 Feb 24. PMID: 33534149; PMCID: PMC8014692.
This manuscript is a resubmission of an earlier submission. The following is a list of the peer review reports and author responses from that submission.
Round 1
Reviewer 1 Report
It would be interesting to know, whether any patient included in this study had a DVT or a PE and if yes, how this influenced the D-Dimer values.
I would also like to have information concerning the DVT prophylaxis. Is this identical in the AJR and spine patients? Does the prophylaxis have an influence on the D-Dimer levels?
Reviewer 2 Report
In the manuscript, the authors stated that the concentration of D-dimers does not differ between SARS-CoV-2 IgG positive and negative patients, who undergo two types of surgical procedures.
The paper contains many shorthands and methodological errors and cannot be published in the JCM.
1. the aim of the study is not clear - why were TKA patients compared with neurosurgery patients? The justification presented by the authors is insufficient.
2. what is the nature of the study - retrospective or prospective?
3. on what basis was D-dimer divided into three categories?
4. D-dimer was measured in plasma 3-21 days prior to surgery. What was the median of this time period? Taking into account the half-life of D-dimer in the blood, this period does not illustrate the actual blood coagulation and fibrinolysis processes in patients.
5. what method was used to determine SARS-CoV-2 IgG titer?
6. the inclusion and exclusion criteria are incomprehensible. Did these criteria apply to both patient groups?
7. what statistical tests were used in the analysis of the results?
8. there is no table summarizing demographic and clinical characteristics of the patients in the results.
9. the comparison of groups of patients differing in such a decidedly size is not correct from the methodological point of view.
10. generally the results are presented very carelessly, without any logical sense.
11. the discussion is trivial and the conclusions of the study are incorrect.
Reviewer 3 Report
Dear Authors, thank you for submitting your paper.
The aim of the present study is to investigate differences in preoperative D-Dimer levels in SARS-CoV-2 IgG positive and negative patients undergoing primary total knee and total hip replacement (TJA) or spine surgery.
I congratulate the authors for this very relevant research, which will add to the dental field.
It appears well structured, correctly carried out and written without logical or factual errors.
Methodological aspects are deeply cleared in the manuscript.
The topic is in line with the journal aim.
-Please introduce in the Materials and Methods section a subparagraph for the statistical analysis.
-Data reported in the Methods section are appropriate and precisely described;.
-I suggest to the Authors to cite in the Introduction the following recent article about the applications of telemedicine and teleorthodontics during COVID-19 pandemic:
https://doi.org/10.3390/jcm9061891
The Conclusions are correctly stated and supported by the findings obtained from the present study.
According to this Reviewer’s consideration, novelty and quality of the paper, publication of the present manuscript is recommended.
Reviewer 4 Report
A pleasantly compact presentation of a currently important issue. Only the presentation of the results could be optimized by adding a further tabular form.
Reviewer 5 Report
Methods and Results
Did you check one-time positive SARS-CoV-2 patients with a second qPCR?
How did you rule out these patients were false-positive? What test was actually used?
Did you perform a power analysis in order to assess the actual number needed in order to find significant differences between SARS-CoV-2 IgG positive and negative patients? Please report the power of your analysis.
Why did you compare SARS-CoV-2 IgG positive spine and TJA patients? It is not surprising that that TJA patients were older.
Regarding the IgG positive and -negative patients in TJA and spine surgeries: What test was performed? Did you assess normal distribution? Or have you performed a non-parametric test?
The more interesting question is: Were the IgG-positive patients significantly younger or had a higher BMI?
Please add a table with mean, sd and n of each patient group with a pairwise comparison and the applied test.
Also: Did you compare the exact values or grouped values?
Discussion
"... are not in a hypercoagulable state ..." - assessed by one parameter? What about rotation thromboelastrography, clotting time, or even aPTT?
The statement in line 117-9: You just showed an association, not a causal connection. Please remove that statement.
Overall: What is the message: Do not measure D-Dimers in SARS-CoV-2 patients? Without a proper follow-up of thromboembolic incidents?
Reviewer 6 Report
Dear Sir,
thank you for giving me the possibility to review the paper “COVID-19 Does
Not Trigger Elevated D-Dimer Levels in Orthopaedic Surgery”.
However, before considering it for publication on JCM, the following issues should be addressed:
1. Please modify the title to highlight the paper focuses on
post-Covid 19 patients (i.e. SARS-CoV-2 IgG positive) undergoing elective
Orthopaedic surgery
2. Page 1: lines 45-46: please explain how D-dimer is useful in the
diagnosis of PJIs
3. Results: please improve data presentation by adding tables and
figures
4. Please add the following pre-op data (if available): patient’s
diagnosis; anticoagulant drugs assumption; PT (INR); aPTT (ratio); white
cells count; neutrophils numbers; CRP; ESR; PCT; presepsin
5. The main drawback of the present study is the lack of information
about the time interval between COVID-19 symptoms resolution and the
surgical procedure. Please provide this datum, otherwise, the study
conclusion might be misleading!
6. Please improve the discussion section by commenting on the
clinical impact of the findings of the present study